# A Review of EMG-, FMG-, and EIT-Based Biosensors and Relevant Human–Machine Interactivities and Biomedical Applications

**DOI:** 10.3390/bios12070516

**Published:** 2022-07-12

**Authors:** Zhuo Zheng, Zinan Wu, Runkun Zhao, Yinghui Ni, Xutian Jing, Shuo Gao

**Affiliations:** School of Instrumentation and Optoelectronic Engineering, Beihang University, Beijing 100191, China; thecoober@buaa.edu.cn (Z.Z.); 19375161@buaa.edu.cn (Z.W.); 19375444@buaa.edu.cn (R.Z.); 19376003@buaa.edu.cn (Y.N.); 19376039@buaa.edu.cn (X.J.)

**Keywords:** FMG, EMG, EIT, biological signal, human–machine interactivities

## Abstract

Wearables developed for human body signal detection receive increasing attention in the current decade. Compared to implantable sensors, wearables are more focused on body motion detection, which can support human–machine interaction (HMI) and biomedical applications. In wearables, electromyography (EMG)-, force myography (FMG)-, and electrical impedance tomography (EIT)-based body information monitoring technologies are broadly presented. In the literature, all of them have been adopted for many similar application scenarios, which easily confuses researchers when they start to explore the area. Hence, in this article, we review the three technologies in detail, from basics including working principles, device architectures, interpretation algorithms, application examples, merits and drawbacks, to state-of-the-art works, challenges remaining to be solved and the outlook of the field. We believe the content in this paper could help readers create a whole image of designing and applying the three technologies in relevant scenarios.

## 1. Introduction

In recent years, with the development of material science and electronic information technology, wearable devices have made great progress. Nowadays, wearable devices can be mainly used in two fields, HMI and medical. Among various wearable technologies, EMG, FMG, and EIT are commonly used to detect biological signals related to nerve and limb movement. When an action occurs, nerves send electrical signals to drive muscles. Then, muscle contraction causes changes in muscle volume and internal impedance. The posture and acceleration will change during the action. The electrical signals can be detected by EMG [1], while the changes in muscle volume can be detected by FMG [2], and internal impedance by EIT [3].

As a technique for detecting electrical activities caused by the muscles, wearable EMG systems are used widely. For instance, J. Qi et al. used EMG technology to recognize different hand gestures, as a result, a long-term recognition accuracy of 79% was achieved [4]. Because EMG detects electrical signals from superficial muscles, its performance is limited by the skin impedance changes caused by sweating and contact [5,6], which cause a decrease in the accuracy of pattern recognition. FMG is an alternative technology that directly captures changes in skin surface pressure due to changes in muscle volume caused by muscle activity [7,8]. Compared to EMG, FMG is robust to electrical interference and sweating, whilst also being non-invasive and inexpensive [9,10]. In the work of Islam et al. [11], the performance of motion detection with FMG and surface electromyography (sEMG) were compared in a daily scenario. They tested four different limb motions in five healthy male subjects. As a result, in one-day training, the day-to-day classification accuracy reaches 84.9% while the accuracy of sEMG reaches 77.8%. However, it is not a simple competition between FMG and EMG [9]. Jiang et al. proposed a novel co-located EMG–FMG sensing armband which can detect FMG signal and EMG signal simultaneously [8]. Five healthy subjects performed gestures of ten American sign language (ASL) digits 0–9. The accuracy of EMG-only gesture recognition was 81.5%, while FMG-only was 80.6%, and co-located EMG–FMG had the best performance of 91.6%. Another potential human–machine interaction technology is EIT. It is an imaging technology that detects the internal structural impedance distribution of objects by external electrical excitation signals. To obtain the internal resistivity of the object, EIT uses electrodes on the boundary to apply a high-frequency alternating signal and measure the response signal. For instance, Zhang et al. [3] designed a wearable hand ring called tomo based on 4-pole EIT, which achieved high accuracy in gesture recognition.

It can be seen from the above examples that these three technologies are applied in similar human-machine interaction and rehabilitation scenarios, so it is necessary to explain and discuss these three technologies in detail. This is beneficial for practitioners to effectively select corresponding technologies in designing specific scenarios, considering their advantages and disadvantages. For this purpose, we wrote this paper to review the development and application of the three techniques of FMG, EMG, and EIT in the past 20 years. We start the review from four parts: principle, hardware, algorithm, and application. First of all, this review summarizes the signal acquisition device and signal processing process of three different techniques. Then, the application of three methods in human–machine interaction and the medical field is mentioned. Finally, we analyze and compare the advantages and disadvantages of the three methods, and then we propose the problems that need to be improved in the future and propose some solutions.

By writing this article, we hope that readers can understand the principles, signal processing processes, application scenarios, advantages, and disadvantages of the three technologies clearly so that subsequent researchers can quickly choose the appropriate technology for research. At the same time, we also hope that researchers can further develop the three techniques to overcome their existing problems. The generation, processing, and application of FMG, EMG, and EIT signals are showed in Figure 1.

## 2. Principle

EMG, FMG, and EIT are emerging methods to obtain human information in recent years. The advantage of the three techniques is that all of them can be measured noninvasively and harmlessly, which means that they have great potential for human-machine interaction. In this section, we will introduce the principles of FMG, EMG, and EIT.

### 2.1. FMG

FMG is an approach to collecting motion signals by sensing changes in muscle volume. Its basic principle is that different muscle activities cause different movements. When an action occurs, the volume of the underlying musculotendinous complex changes, which results in a change in the distribution of surface mechanical forces. Different movements are encoded into different force images. By decoding these images, original motion information can be obtained, which has been widely used in gesture recognition [2], human–machine collaboration [16], prosthetic control [17], and operational force estimation [12].

Generally, researchers can use force sensors matrix/array to detect the mechanical force in the FMG technique [18]. The force sensor reflects the magnitude of the force applied to the sensor. When a socket with many sensors is wrapped around a part of the limb, the muscle force map can be obtained. With some algorithms, such as machine learning [9], the original motion information (type of movement and magnitude of force) can be obtained by using the FMG signal. An example of FMG signal output is shown in Figure 2.

### 2.2. EMG

EMG refers to a series of electrical signals associated with muscles due to neurological control and generated during muscle contraction. This signal is generally given by the experimental method, which can represent the physiological characteristics of muscles after amplification and processing [19,20].

EMG is derived from the brain to muscle control. It is based on three steps: resting potential, depolarization, and repolarization. Its formation is caused by the concentration difference of Na^+^ ions, K^+^ ions, and Cl^−^ ions, but it is dominated by Na^+^ ions. When the muscle does not contract, the concentration of Na+ ions in muscle cells is greater than that out of muscle cells. With the ion pump, Na+ ions outflow forms a resting potential with positive external potential and negative internal potential on the membrane of muscle fiber. For example, when trying to move upper limbs, our brain sends movement control signals to the muscles, which are transmitted to the muscles through the nervous system. When the signal reaches the muscle fibers, chemicals such as acetylcholine are released at the nerve end, causing a large influx of Na^+^ ions, which rapidly form an action potential in the muscle fiber, a process known as depolarization. After the signal transmission, with the action ion pump, muscle fibers quickly return to the state of resting potential, which is called repolarization. The combination of all the muscles’ action potentials of a motor unit is called a motor unit action potential (MUAP) [21]. The superposition of MUAP in space and time produces EMG. The EMG signal generation process is shown in Figure 3.

### 2.3. EIT

EIT is an imaging technology that detects the internal structural impedance distribution of objects by external electrical excitation signals. By placing a set of electrodes on the surface of the conductive object to be measured, EIT applies a high-frequency alternating current to each electrode pair as the excitation signal and measures the electrical response signal on other electrode pairs in turn to obtain the internal resistivity of the object. Due to its advantages of non-radiation, non-damage, low cost, and simple structure, EIT has been widely used in non-destructive testing, geological exploration, and other fields. Nowadays, the application of EIT in biomedical imaging and human–machine interaction has been widely studied. 

The human body is a complex structure with different electrical impedance distributions. There has been a lot of research on electrophysiology, which is concerned with the electrical properties of biological tissues, and the principle of them is very complex and influenced by frequency, temperature, and direction. This is closely related to the structure and function of the tissues. Generally speaking, the blood and muscle with high extracellular water content and electrolyte concentration have a relatively low electrical impedance. In contrast, fat, bone, and air increase impedance. This difference gives each tissue and state certain characteristics. For organisms, when controlling the amplitude and frequency of excitation signals within a safe range, the output signal and calculate impedance distributions can be harmlessly measured.

The impedance characteristics of organisms often change in certain situations. For example, the electrical impedance of the lungs depends to a large extent on the concentration of the internal air. When air is inhaled, the electrical conductivity of lung tissue concomitantly decreases. The flow or clotting of blood also causes impedance changes. When the body tissue is diseased, its electrical impedance may change significantly, which will be detected by EIT, to be applied to medical diagnosis and treatment. Similarly, the limbs in different postures also correspond to different impedance distributions. Therefore, the impedance distribution of the part of the body can be measured by EIT to realize posture detection.

According to the different imaging purposes, EIT can be divided into two types: static imaging and dynamic imaging. Static imaging calculates the absolute value of impedance distribution and has a wider range of applications. However, it is more computationally intensive and vulnerable to noise, resulting in low image resolution. In contrast, dynamic imaging computes the relative impedance distribution and produces a differential image, which suppresses noise very well. Depending on the measurement method, it can be further divided into time difference imaging technique and frequency difference imaging technique. Time difference imaging obtains the difference of impedance at two times, while frequency difference imaging obtains the difference of impedance at different frequencies at the same time. Dynamic imaging is less affected by noise and relatively simple to calculate, but it is essential to ensure that impedance changes exist, so the application is constrained. EIT signal acquisition and reconstruction are shown in Figure 4.

## 3. Data Acquisition

In this section, we will introduce the signal acquisition methods of FMG, EMG, and EIT. We successively introduced the sensors used for FMG, the sampling frequency and channel number configuration, the EMG sampling method and electrode type, the sampling frequency, and channel number configuration, and finally, we introduced the electrode configuration of EIT and the drive pattern.

### 3.1. FMG Signal Acquisition

FMG technique uses force sensors to obtain information on the underlying musculotendinous complex changes during movements [7]. There are many types of force sensors used in FMG, for instance, piezoresistive- [22], capacitive- [23], piezoelectric- [24], optoelectronic- [25] and pneumatic-based [26] sensors.

#### 3.1.1. Piezoresistive Sensors

To acquire effective biosignals, the sensor needs to be in close contact with the skin, and piezoresistive sensors have this characteristic. The most frequently used piezoresistive sensors are force-sensitive resistors (FSR), for instance, FSR 402 [27,28,29] and FSR 400 [16,30], which are based on resistive polymer thick film sensor (RPTF) technology. Because of their thin profile, flexibility, and low cost, they become a practical solution for prosthetic pressure measurement [31].

The structure of FSR 400 series is often composed of two layers, one is the printed semiconductor layer on the bottom layer and the other is the interdigitating electrode on the semiconductor layer. When pressure applied to the active area increases, the resistance values of the piezoresistive material will decrease. The force sensitivity range of FSR 400 is 0.2 N–20 N, and its hysteresis is 10% [32].

The advantage of the piezoresistive sensor is its simple structure and affordability, but it suffers from heating issues and high hysteresis [33].

#### 3.1.2. Capacitive Sensors

Capacitive sensors are another sensor used to detect FMG signals [23,34]. The capacitive sensor reflects the force/pressure loaded on it by detecting the capacitance value of capacitance. To achieve this, an elastic material between two electric layers is necessary. When the pressure applied to the sensor changes, the distance between two electric layers changes, resulting in a change in the capacitance value of the sensor [35].

Polydimethylsiloxane (PDMS) is a frequently used material in dielectric layers. Lei et al. used PDMS as the dielectric layer in a 16:1 mix ratio. The sensor can measure the pressure up to 945 kPa, and obtain a high sensitivity of 6.8%/N [35]. Maddipatla et al. used silver (Ag) ink on a flexible polyethylene terephthalate (PET) as electrodes, and a 16:1 mixing ratio of PDMS as a dielectric layer to fabricate a force sensor. The sensor offered a sensitivity of 0.13%/N from 0 N to 10 N [36].

Capacitive sensors have the advantage of low power consumption and fast response, but they are sensitive to electromagnetic interference (EMI) noise and are not suitable for long-term use.

#### 3.1.3. Piezoelectric Sensors

Piezoelectric sensors have good dynamic force-sensing performance. When pressure is applied to the sensor, a potential difference is generated between the upper and lower plates of the sensor. By measuring its voltage, the magnitude of the pressure can be obtained. Common piezoelectric materials can be divided into ceramics, films, and fibers [33].

The acquisition of human biological information places high demands on the flexibility of sensors. However, it is difficult to achieve good flexibility for piezoelectric sensors based on ceramic materials such as silicon, PZT, and glass. Therefore, flexible piezoelectric sensors based on polyvinylidene fluoride (PDVF) are widely used in FMG. For instance, in Chuang et al., a PVDF-based tactile sensor is proposed, which exhibits good linearity from 0.5 N to 4 N with a sensitivity of 6.4 mV/N [37].

The advantage of piezoelectric sensors is low hysteresis, strong sensitivity, and low power consumption. However, due to their characteristics, piezoelectric sensors cannot be used in static force sensing.

#### 3.1.4. Other Sensors

In addition to the above sensors, there are some less commonly used sensors as follows:

Optical fiber sensor is an emerging pressure measurement sensor in recent years, which is mainly composed of multimode fiber. When force or displacement is applied to the sensor, it causes the flexible layers above and below the fiber to bend. The core propagating modes and the cladding radiation modes will be coupled because of the consecutive variations on the bending radius [38], which will cause light attenuation. By sensing the degree of light intensity attenuation, the applied force or displacement information can be obtained.

Fujiwara et al. proposed a low-cost optical fiber force myography sensor [2]. According to Fujiwara, this sensor has very excellent performance, with a sensitivity of 0.05 N and a large range of 0–22 N. At the same time, the sensor has excellent linearity between the normalized light intensity and static pressure in static force measurement (the correlation coefficient is 0.98). Compared with piezoresistive sensors, fiber optic sensors have smaller linear errors and lower latency, but higher energy consumption and lower spatial resolution [33].

Pneumatic sensor is an early type of force sensor in FMG. As early as 1999, Abboudi et al. used tendon-activated pneumatic (TAP) sensors to control three-finger prosthetics, whose sensors were fabricated from porous polyurethane foam and vacuum-formed within a polyethylene bag [26]. For pneumatic sensors, it is difficult to ensure that the internal pressure remains stable under external temperature changes, and the gas also has leakage problems. Therefore, pneumatic sensor is an early sensor used in FMGs and has been abandoned by researchers in recent years.

#### 3.1.5. Sampling Frequency and Channels

To obtain more information about the movement, more channels are necessary. Lei et al. investigated the best configuration of sampling frequency and channel numbers in FMG. They proposed that the number of channels greatly affects the accuracy of gesture recognition. The more channels, the higher the recognition accuracy. At the same time, they recommend using eight channels in the field of gesture recognition, which can achieve a satisfactory result [39]. 

The value of the sampling frequency should be adjusted according to the target signal. Low sampling frequency will filter out important information, which will eventually reduce the recognition effect; high sampling frequency will introduce noise and increase the cost of subsequent signal processing devices. Therefore, choosing a suitable signal sampling frequency can not only obtain better experimental results but also reduce the cost of the entire system. 

Xiong et al. investigated the frequency of different hand postures of people. After their research, they proposed that the frequency of human hand movement is mainly located in the low-frequency region below 10 Hz [40]. Therefore, finding the highest frequency which can reflect the complete motion information has great significance in reducing costs. Many teams are searching for the most suitable sampling frequency. In upper limb movement recognition, Zhen et al. used a frequency range of 1–500 Hz to collect FMG signals from the wrist and the bulk region of the forearm of 12 participants for movement recognition [41]. Finally, they propose that the minimum sampling frequency when researchers capture the FMG signals from the forearm and wrist is 54 Hz and 58 Hz for distinguishing static actions, and 70 Hz and 84 Hz for distinguishing dynamic actions.

In practical applications, the FMG signal does not have a clear requirement for the sampling frequency and the number of channels, and different configurations have different performances in different applications. In the field of gait recognition, Jiang et al. used FSR 402 to recognize four gaits using eight channels at a sampling frequency of 500 Hz. They obtained an overall sample-based accuracy of 91.3% ± 3.3% [28]. Sakr et al. used FSR 402 to estimate hand force using 16 channels at a 10 Hz sampling frequency. Finally, they obtained the *R*^2^ accuracies of 0.83 for the 3-DoF force, 0.84 for 3-DoF torque, and 0.77 for the combination of force and torque (6-DoF) in cross-trial evaluation [42]. Except for the most widely used FSR sensor, the sampling frequency of other sensors is different, but most of them choose a lower sampling frequency to match the human body movement. Wu et al. use two fiber optic sensors to recognize gestures at a sampling frequency of 100 Hz; they get an average precision and accuracy of ~99% and ~99.8% [25]. Zakia et al. used 16-channel TPE 502C to predict human hand force at a sampling frequency of 50 Hz, and the force estimation accuracy under 1-DOF is 90–94% [12]. A summary of the four commonly used sensor technologies is shown in Table 1.

### 3.2. EMG Signal Acquisition

#### 3.2.1. Acquisition Method

The collection methods of EMG mainly include the invasive technique of inserting needles into muscle tissue and the non-invasive technique of skin surface sampling. The EMG collected by these two methods is called invasive electromyography (iEMG) and surface electromyography (sEMG), respectively. To collect the iEMG signal, the electrodes are needle-shaped and inserted into the subject’s muscles. Firstly, the process of collecting iEMG requires the subject to be anesthetized, which will bring pain to the subject and easily cause infection. The whole process is harmful to the health of the subject to a certain extent. Secondly, the acquisition of iEMG is cumbersome and the cost is too high. In contrast, sEMG is collected by sticking electrodes on the subject’s skin. It overcomes many shortcomings of iEMG. It has the advantages of being painless, low infection risk, low cost (compared with iEMG), and a simple collection process. So, it is more suitable for EMG sampling [19,49].

#### 3.2.2. Electrode Profile

As we analyzed in the previous section, sEMG is more commonly used in practice than iEMG. So, we emphatically discuss the electrodes of sEMG in this section. In contrast, sEMG acquisition relies on electrodes stuck to the skin surface, so the electrodes are crucial to the quality of EMG. Electrodes can be divided into wet electrodes and dry electrodes. Their materials, characteristics, and application scenarios are quite different. Their comparison is shown in Table 2.

For the electrode distribution, it can be divided into two categories: the method based on sparse multi-channel EMG and the method based on high-density EMG. The former method requires accurate pasting of the electrodes to the target muscles due to the small number of electrodes and their independent existence. The latter method typically needs two-dimensional electrode arrays that can cover multiple muscles within a single skin. It can reflect changes in EMG over time and space [57,58,59]. The model of the wet and dry electrodes is shown in Figure 5.

#### 3.2.3. Sampling Frequency and Channels

The acquisition of EMG signal usually has the following indicators: the number of channels, the sampling frequency, and so on. The number of channels reflects the number of muscles collected. The sampling frequency is reflected in the time dimension. An appropriate increase in the sampling frequency can more effectively reflect the mutation situation of muscle movement. 

Channel number, sampling frequency, and noise rejection performances are the key indexes to measure the quality of EMG acquisition system. For the number of channels, when the action is simple, the sensor with few channels can be used. For complex movements or the combination of many single movements, it is appropriate to use multi-channel electrodes. For sampling frequency, when we just need to detect the same simple event, such as tremor or any on–off event, a low sampling frequency can accomplish the task. However, if we need to focus on the details of one movement in order to better separate it from the others, we need a high sampling frequency in that case. Generally, the sampling frequency of each channel is usually the highest frequency of the recorded EMG signal to satisfy the Shannon sampling theorem. At the same time, the multi-channel acquisition system samples all channels synchronously, so that no phase error is introduced.

If the action is simple, for instance, open and close, a three-channel or four-channel sensor and low sampling frequency can be used in order to save the cost of acquisition, which is widely used in the lower limb movement acquisition. Zhou et al. used three-channel EMG sensors to identify lower limb movements [60]. They use a three-channel EMG sensor with bipolar electrodes to collect lower limb movement data at a frequency of 200 Hz. After amplification and filtering, high-quality EMG signals are finally output. However, if the sampling action is multiple and complex, the sensor with more channels and high sampling frequency is needed in order to increase the number of simultaneous muscle sampling. Six-channel sensors and eight-channel sensors are commonly used multi-channel sensors. They are widely used in upper limb motion acquisition. For example, in the field of gesture recognition, Hand Open, Wrist Flexion, Wrist Extension, Radial Deviation, Ulna Deviation, Forearm Pronation., Forearm Supination, and Hand Closure are commonly used gesture sets of eight-channel sensors. Eight-channel sensors and their combination can cover all hand movements in a relatively comprehensive way [61].

Kanoga et al. used an eight-channel sensor to collect data on the right upper limb at a sampling frequency of 200 Hz for prosthesis control [62]. Melissa L. B. Freitas et al. used an eight-channel bipolar armband at a sampling frequency of 2000 Hz, with electrodes placed on the forearm and reference electrodes placed near the ankle. Six gestures: wrist flexion, wrist extension, right wrist flexion, left wrist extension, forearm supination, and forearm pronation, were collected from 13 volunteers and repeated 300 times, generating a dataset containing 3900 motion sequences [63].

### 3.3. EIT Signal Acquisition

#### 3.3.1. Electrodes

For anthropometry, EIT usually uses a set of surface electrodes as the injection and measurement electrodes, and the performance of the electrodes directly determines the quality of the data collected. Ideal EIT electrodes should have high conductivity, stable performance, a simple process, and low cost. Common electrode shapes include round, rectangular, and composite electrodes with a specific number, such as 8, 16, 32, etc. Common electrode materials include copper, silver, and titanium, which have much higher conductivity than human tissues. Yao et al. [64]. studied the influence of different kinds of electrodes on EIT detection and classification accuracy, which includes copper electrodes, conductive cloth electrodes, and medical electrodes. Their results show that the rectangular copper electrode has the highest accuracy.

The quality of images is important to EIT, and researchers have adapted many approaches to improve the sensors in order to get a better resolution and accuracy of the images. The most direct way is increasing the number of electrodes [65,66], which enables more data to be generated for image reconstruction. Increasing the number of electrodes was found to have a significant impact on the image quality; however, Wang et al. have used FEM to prove that when the electrodes increase to a certain amount, the gain of increasing numbers will be less significant [67]. In addition, more electrodes may cause practical problems in data collecting and electrode placing [65].

#### 3.3.2. Drive Pattern

The most common way is to use the current drive and voltage measurement nowadays. When the device works, the electrodes inject excitation current in turn, with a frequency in the beta range (10 kHz–1 MHz), which has been proved to be best suited for the measurement of tissue impedance. Then, other certain electrodes measure the voltage response and pass the data to the backend. In modern EIT systems, the frequency of injection current is usually varied in a range between 50 kHz and 250 kHz. The maximal root mean square of the injection current is regulated in the standard IEC 60601-1 [68]:$$i_{RMS,\max}\left(f\right)=\left\{ \begin{array}{l} 100\mu A\;f\leq 1\mathrm{kHz}\\ 100\mu \cdot \frac{f}{1\mathrm{kHz}}\;1\mathrm{kHz}<f<100\mathrm{kHz}\\ 10\mathrm{mA}\;f\geq 100\mathrm{kHz} \end{array}\right.$$

Choosing different electrode pairs for injection and measurement will lead to different driving modes in the EIT system, so as to obtain different effects. Therefore, researchers have explored and implemented different driving patterns in the EIT system. The traditional driving methods include adjacent driving, opposite driving, cross driving, adaptive driving [69,70,71], etc. Adjacent driving uses two adjacent electrodes to inject excitation, with better stability and measurement accuracy; opposite driving uses opposite electrodes and has better resolution; in the cross driving, two driving electrodes are placed at 90° and their voltages are measured on adjacent positions. Their performance is intermediate between adjacent driving and opposite driving [72].

The EIT systems can be divided into two categories of working modes including four-terminal and two-terminal. Generally, the four-terminal mode can reduce the impact of contact impedance, which makes it widely used in many situations as gesture recognition. However, this mode is more complicated, and its applicable range of excitation frequency is narrow. Conversely, the two-terminal mode has a low cost and a wider application range of excitation frequency. Lu et al. have proposed a two-to-four-terminal mapping algorithm, eliminating the contact impedance of the two-terminal mode. Their work also showed that the gesture recognition rate of four-terminal EIT is lower than two-terminal, which proved that contact impedance is important in gesture recognition [73].

## 4. Data Processing

The original signal obtained by sensors cannot be used directly. In order to apply the collected signals to practice, we need to apply some signal processing steps, such as filtering and feature extraction, and then use algorithms, such as machine learning, to connect the original signals with practical applications. In this section, we will introduce some data processing methods for the three signals. All steps of data processing are shown in Figure 6.

### 4.1. Data Preprocessing

In the preprocessing stage, we mainly filter and amplify the signal. At the same time, for different signals, there are their own unique signal preprocessing methods, which will be mentioned in other preprocessing.

#### 4.1.1. Filter and Amplification


**Filter and amplification of FMG**


Because the frequency of human motion is mostly below 10 Hz [40], researchers usually choose a low-pass filter to process the signal preliminarily. The most frequently used low-pass filter is the 4th and 5th Butterworth filter. Amit Kumar et al. adopted a low-pass filter using the Butterworth (5th order) filter with a cut-off frequency of 10 Hz to filter out high-frequency components in FSR [30]. While in optoelectronic sensor, Wu et al. used a 4th order digital Butterworth low-pass filter with the cutoff frequency set at 200 Hz [25]. In order to make the signal more intuitive, some researchers also adopted a high-pass filter to filter out the linear trend of the FMG signal. Sakr et al. adopted a high-pass 5th Butterworth filter with a cut-off frequency of 0.5 Hz to remove the DC component.

The magnitude of the FMG signal is positively correlated with changes in muscle volume. Generally speaking, the muscle volume changes significantly during exercise, so the amplitude of the FMG signal is large enough, and subsequent amplification is not required.


**Filter and amplification of EMG**


In EMG, the upper cut-off frequency of the filter is generally less than 500 Hz, while the lower cut-off frequency is generally more than 20 Hz [19]. When filtering, Butterworth filters are often applied. Kanoga collected signals from the right upper limb of the subjects and performed high-pass filtering at a frequency of 15 Hz through the fifth-order Butterworth filter [62]. Another effective interference filtering method is wavelet transform, which can effectively decompose continuous signals. Wavelet transform has many advantages, such as short sampling time, ease to avoid unwanted signals, and can analyze signals with much missing information [20,74]. The selection of wavelet is a key part of wavelet denoising, and the selection indexes can be roughly divided into the type of wavelet function, the scale, and the threshold [20]. Karan Veer et al. used wavelet transform to interpret surface electromechanical signal diagrams for the classification of upper arm movements. They collected the upper limb movement information of 10 amputation volunteers and selected The Daubechies wavelet family for signal processing of EMG, with the final motion classification accuracy reaching more than 85% [74].

EMG’s amplitude is quite small. When the muscle does not contract, the amplitude of the EMG signal is generally in the range of −80~–90 mV. However, when the muscle contracts, the amplitude is only a few hundred millivolts at most [75]. So, in order to acquire an observable signal, the EMG signal is often amplified by 50~100 times to reach above 1~2 volt.


**Filter and amplification of EIT**


The frequency of the output signal of the EIT depends on the excitation signal. Therefore, a filter is needed to obtain the signal of the desired frequency band based on the frequency of the excitation signal. At present, the commonly used impedance measurement chips often integrate numerous modules, including filters. For example, AD5933 has an LPF inside, and ADS1256 has a sinc5 filter. These are usually programmable devices. When used, the cut-off frequency can be selected according to the set excitation frequency. Moreover, digital filters are also used for filtering. In addition, electromagnetic interference in the environment is also a problem to be considered, which usually requires a low-pass filter to eliminate.

Traditional filters usually set a fixed cut-off frequency to eliminate noise in a specific frequency band, which is not very effective for time-varying random noise suppression. Therefore, researchers have proposed filtering algorithms based on statistical methods, such as the Kalman filter and adaptive filter. At present, these algorithms have been applied to EIT and achieved good results. For example, Baidillah et al. [76]. Proposed implementation of adaptive noise cancellation (ANC) algorithms, which are least mean square (LMS) and normalized least mean square (NLMS), filter onto a field-programmable gate array (FPGA)-based EIT system that effectively improved the imaging quality of EIT.

The EIT applied to human signal detection requires its excitation current to be controlled in a safe range (usually less than 10mA), so the output signal is small and must be amplified. Common amplifiers include instrument amplifiers and programmed gain amplifiers.

#### 4.1.2. Other Preprocessing


**Normalization of FMG**


The signal was captured under different circumstances, which means that researchers could not use the same method to analyze all signals. Therefore, the standardized part is very important. At the same time, standardization can minimize the error caused by installation. Usually, the following formula is used to normalize the signal:(1)$$x_{norm}=\frac{x_{ori}-\min\left(x_{ori}\right)}{\max\left(x_{ori}\right)-\min\left(x_{ori}\right)}$$
where, $x_{norm}$ and $x_{ori}$ represent the data after normalizing and the original data, respectively.


**Data segmentation of EMG**


EMG signals obtained by preprocessing cannot be used for feature extraction directly. In order to separate the useful parts, it is necessary to add windows. Window addition technology can be divided into overlapping windows and adjacent windows. In the overlapping window technique, the two contiguous windows overlap, but in the adjacent window technique, the two contiguous windows have a time gap. The main parameter affecting the performance of data segmentation is the window length. The window length determines the amount of data and useful information. The longer the window length is, the more useful information is, and the more accurate the prediction will be, but the system will take more time to handle it [77]. Data segmentation is a relatively rough extraction process, which helps to estimate the expected results of the EMG classifier while separating the interference information [78].


**Phase-sensitive demodulation of EIT**


Since EIT technology often uses multi-frequency EIT to obtain rich complex impedance information, a phase-sensitive demodulator (PSD) is used to separate the real part and the imaginary part. PSD includes analog demodulation and digital demodulation. Common analog demodulations include switch demodulation, analog multiplier demodulation, and zero-crossing phase discriminator demodulation. Digital demodulation uses high-performance digital devices, such as FPGA and DSP, to extract the amplitude and phase information of the measured signal by numerical method. Common demodulation methods include FFT demodulation and orthogonal sequence demodulation. Chen et al. have designed a new type of dual-frequency PSD impedance measurement circuit that can measure both resistance and capacitance with an uncertainty of less than 0.5% [79]. Ge et al. have proposed a novel FPGA-based digital PSD for EIT to obtain a signal with high SNR, high data precision, and high demodulation speed, and the emulation results showed that the phase error is 1.03 [80].

### 4.2. Feature Extraction

In order to use the original signal in the algorithm, we need to perform feature extraction on the preprocessed signal. Feature extraction is aimed at further screening out useful signals that can reflect the movement’s feature. For FMG and EMG, the features are mainly extracted in the time domain (TD), frequency domain (FD), and time–frequency domain (TFD). For EIT, we generally do not do redundant processing, but directly extract its magnitude as a feature.

#### 4.2.1. Features of FMG and EMG

The time-domain characteristic represents the change of signal amplitude with time, which can be obtained directly from the original signal without further signal transformation. Frequency domain characteristics include the power spectral density of the signal, which need to be transformed into the frequency domain to obtain them. The combined signal characteristics in the time domain and frequency domain are defined as the signal characteristics in the time–frequency domain [81].

In time domain, there are five commonly used parameters to describe signal features: mean absolute value (MAV) [17,82,83], mean absolute value slope [22,27], slope sign changes (SSC) [83,84,85], zero crossing (ZC) [83,84] and waveform length (WL) [86]. These features mainly apply to signal amplitude, while in the frequency domain, signal power information and frequency information are mainly utilized. There are also some parameters to describe signals in the frequency domain, such as mean frequency (MNF), mean power (MNP), power spectrum ratio (PSR), and so on [87].

In the time–frequency domain the information used for feature extraction is more abundant, but it also brings the problem of complicated calculations and long processing time. Therefore, many fast algorithms are applied. In recent years, the frequency of wavelet transforms and wavelet packet transform used for feature extraction has gradually increased. The accuracy with which they interpret signals is remarkable. Among them, the former has low computational complexity, while the latter can process both high-frequency and low-frequency components [82].

The parameters commonly used in each domain are summarized in Table 3 [87]:

#### 4.2.2. Feature of EIT

Unlike EMG and FMG, EIT takes boundary information (voltage or impedance values) collected as a feature of reconstructed images and machine learning. To avoid dimension explosion, it is sometimes necessary to reduce the dimension of data by selecting a feature subset with a higher weight. Ma et al. [88] demonstrated the feasibility of finding a subset of features in the original data of EIT to reduce the measurement time and maintain acceptable accuracy at the same time. They computed the SHAP (Shapley additive explanations) values of the dataset and found the most valuable electrode pairs. Then they established a map between features and electrode pairs, and then obtain a simplified drive pattern with only part of the whole electrode pairs. This way helped them to reduce 50% electrode pairs, which meant half of the measurement time was saved.

### 4.3. Interpretation Algorithms

After all of the signal processing steps are finished, the most important part is the construction of the application algorithm. In the early days, limited by poor computer capabilities, early algorithms were mainly based on mathematical models, using pure mathematical equations to solve signals. At present, with the promotion of machine learning, more and more researchers have begun to use machine learning algorithms to process signals, and get better results.

#### 4.3.1. Early Algorithms


**Vector Decoding and Threshold-based Classification of FMG**


In 2001, Curcie et al. proposed a pressure vector decoding method (PVD) to decode finger commands [89]. They use eight myo-pneumatic (M-P) sensors in the socket to obtain the pressure around the flexor digitorum superficialis and the flexor carpi ulnaris.

The main idea of PVD is to derive a matrix of signal features from a P-dimensional (P-D) pressure vector, and take the pseudoinverse of the feature matrix as a filter (weighting) vector, to obtain a control vector, *Y*(*i*)_*f*_, for each of N fingers (*f* = 1…N). *Y*(*i*)_*f*_ is calculated as
(2)$$Y{\left(i\right)}_{f}={\left|\vec{s\left(i\right)}\cdot \vec{w_{f}}\right|}^{2}={\left|\sum_{k=1}^{P}\left(s{\left(i\right)}_{k}\cdot w_{f,k}\right)\right|}^{2}$$

Here, $\vec{s\left(i\right)}$ is the pressure vector and $\vec{w_{f}}$ is filter vector. To obtain a filter (weight) vector, they choose RMS amplitude of a signal sequence during repeated specified movements to comprise a *P*
*×*
*N* feature matrix **H**:(3)$$H=\left|\begin{array}{cccc} x_{1,1} & x_{1,2} & \cdots & x_{1,N}\\ x_{2,1} & x_{2,2} & \cdots & x_{2,N}\\ \vdots & & \cdots & \vdots \\ x_{P,1} & x_{P,2} & \cdots & x_{P,N} \end{array}\right|$$

Here, $x_{P,N}$ means the RMS of myo-pneumatic sensor *P* when the finger *N* moves. The filter **W** is the discrimination matrix
(4)$$W=H^{-1}=\left|\begin{array}{cccc} w_{1,1} & w_{1,2} & \cdots & w_{1,P}\\ w_{2,1} & w_{2,2} & \cdots & w_{2,P}\\ \vdots & & \cdots & \vdots \\ w_{N,1} & w_{N,2} & \cdots & w_{N,P} \end{array}\right|$$

Each of the rows of **W** is a filter (weighting) vector, $\vec{w_{f}}$. It means the relative contribution of each input to the corresponding output.

The PVD method connects input and output by using artificially set parameters, it struggles when faced with complex situations. In Curcie’s work, they used PVD to recognize the tapping of three fingers, which has great limitations in real-life conditions.

The threshold-based classification method is another traditional algorithm. Because the signal obtained by FMG is the muscle surface pressure signal, which reflects the change of muscle contraction, its amplitude contains movement information to a certain extent. In consideration of simplifying the procedure, Muhammad chose slope to displace amplitude. They set four thresholds to distinguish four hand motions: rest, open, close, and grasp [27].

However, threshold-based classification and PVD methods can only distinguish simple motions. When it is faced with more delicate information, such as different gestures and forces, it is difficult to obtain satisfactory results using these methods.


**Pure Mathematical Theory of EMG**


Traditional methods of EMG classification are almost based on pure mathematical theory. Statistical theory is among the most commonly used theory. For example, in 2004, Raphisak et al. used the sliding window method to calculate the noise interference. They applied a parameter called “moving variance” to calculate the fluctuation of EMG signals, which indirectly evaluated the degree of interference. The moving variance can be calculated with the formula below:(5)$$m_{1}\left(i\right)=m_{1}\left(i-1\right)-\frac{x\left(i-W_{2}-1\right)+x\left(i+W_{2}\right)}{W}$$
(6)$$m_{2}\left(i\right)=m_{2}\left(i-1\right)-\frac{x{\left(i-W_{2}-1\right)}^{2}+x\left(i+W_{2}\right)}{W}$$
(7)$$v\left(i\right)=m_{2}\left(i\right)-m_{1}{\left(i\right)}^{2}$$

Here, $i=W_{2}+2,\ldots ,W_{2}-L$ The length of origin EMG signal is *L*. The sliding window stretched out the data by $W_{2}$ samples on each side. So, the total length of the data is $W=2W_{2}+1$. *v*(*i*) is the moving variance [90].

In 2002, Kilner et al. used a statistical method called “Blind Signal Separation” in order to eliminate the correlational component of crosstalk when recording EMG signals. The “Blind Signal Separation” algorithm used an unmixing filter, which was used on the interfered data. This algorithm produced the EMG signals whose crosstalk was cut down [91].


**Reconstruction of EIT**


To reconstruct an image of the impedance distribution, the researchers first needed to build a mathematical model of the EIT. According to Maxwell’s equations, the mathematical model of EIT can be constructed, including the EIT forward problem and inverse problem. The EIT forward problem refers to solving the boundary potential of the field by finding the conductivity distribution and electrical excitation signal of the measured field, through which to obtain the sensitivity matrix of the boundary measured voltage data to the change of conductivity at different positions in the measured area. This can be solved by the finite volume method (FVM), finite difference method (FDM), and finite element method (FEM). The EIT inverse problem is deriving the internal impedance distribution information according to the measured voltage at the given boundary. Assuming that the initial conductivity distribution is *σ*0, the problem can be described as:(8)δV≈Jδσ+w
where, *δV* is the potential difference and *δσ* is the conductivity difference. J is the sensitivity matrix, which is a Jacobian calculated in the forward problem, and w is a noise vector. This is a mathematically ill-posed non-linear inverse problem, which means that the minimum error and noise in the measurement process may have a great impact on the final result. To solve this problem, researchers usually linearize the non-linear problem and use regularization to simplify the ill-posed problem, and finally obtain the numerical solution by an iterative method. Common regularization methods include truncated SVD, Tikhonov regularization, and Newton’s one-step error regularization.

There are two main categories of methods to solve the inverse problem: iterative methods and non-iterative methods. Generally, iterative methods solve an optimization problem to minimize the difference between the measured signal (such as voltages) and the data predicted by the forward model in each step. Then, the forward model will be tuned to produce data for each iteration until the end of the iteration. A common method is the Newton–Raphson method, which is a variation of Newton’s iteration method and can effectively obtain the numerical solution of equations.

Non-iterative methods are mainly included of non-iterative linearized methods and methods that solve the full nonlinear problem without iterations. Researchers have proposed some commonly used non-iterative method, such as Bayesian inversion, Factorization methods, and D-bar methods and so on [*]. The back-projection method is also a common method, which is a dynamic imaging method that was very successful for simple two-dimensional geometries. The current floods a region from source to drain. By applying excitation in turn and tracing the current to get the equipotential lines between each pair of electrodes, the impedance distribution can be obtained from their superposition [88]. This algorithm is rough and accompanied by star artifacts. However, because of its simple calculation and fast reconstruction speed, it is still used today.

#### 4.3.2. Machine Learning

Machine learning is a powerful tool for digging into information. Nowadays, machine learning is widely used in FMG, EMG, and EIT. It can be divided into two categories, one is based on classification, which can classify discrete states, such as gesture recognition, and another one is based on regression, which can predict continuous parameters, such as force/torque estimation [9,19,87].

The most frequently used strategies in classification are linear discriminant analysis (LDA), support vector machine (SVM), and artificial neural network (ANN) [9]. LDA is the most popular algorithm because of its simple principle. The main idea of LDA is to project data from a high-dimensional space into a lower-dimensional space. It ensures that the variance within the classes of each category is small and the mean between classes is large. In the work of Jiang et al., they used the LDA algorithm to fuse FMG and Leap Motion for virtual gesture recognition [92]. Six grasps gestures of 11 subjects were collected in the experiment, and as a result they got a grasp classification accuracy of 93.4% by using LDA. Because LDA can be easily applied in real-time, it is widely used in three techniques, for instance, Xiao et al. [93], Godiyal et al. [94], Zhang et al. [95], and Huang et al. [96].

The main idea of SVM is to use a hyperplane to divide the data into two classes to maximize the separation between them [97]. If the training data can be ideally separated by a hyperplane (called linearly separable), what SVM needs to do is to optimize the hyperplane parameters to maximize the data gap between different classes. Even if the data are linearly inseparable, SVM can obtain good classification results under the condition of allowing a certain error. The key factor affecting the effect of SVM is the selection of kernel function and parameter values. Subasi [98] used particle swarm optimization (PSO) to optimize the SVM parameters for the diagnosis of neuromuscular disorders in EMG classification. He got an accuracy of 97.41% on 1200 EMG signals collected from 27 subjects. Ha et al. [99] explored the prediction of four hand gestures using FMG and piezoelectric sensors. Classification accuracy of 80% has been achieved through the SVM classification algorithm. Many other researchers choose the SVM algorithm in their works, for instance, Zakia et al. [100], Belyea et al. [101] McDermott et al. [102], and Alkan et al. [103].

ANN is another effective algorithm in machine learning technology. It is an information manager model of biological nervous system functions similar to the human brain [104]. In ANN algorithm, a complex neurons network is constructed with a large number of processing units in different layers. The unit can accept the information of the previous unit and pass it to the next unit after processing. Output the final result after multiple layers of processing. The biggest advantage of ANN is that it is easy to use and can handle multiple input problems [104]. It can be used in many fields. For example, Yap et al. implemented ANN to classify four hand motions through FMG and showed a real-time accuracy of 95% [105]. In the EMG technique, Ahsan et al. used ANN to detect hand movements for human–machine interaction with an average success rate of 88.4% [106].

There are some other machine learning algorithms used in the past few years. Chegani et al. [107] used a random forest regression algorithm to achieve fine finger regression by FMG. In their work, an average squared correlation coefficient of 75% was gained, which showed that it was feasible for FMG to predict finger movements. In Al-Faiz et al.’s [108] work, a *k*-nearest neighbor algorithm was closed for human arm movement recognition. The results showed a good performance in classification to lower signal-to-noise ratio signals. To be brief, machine learning provides a convenient tool to implement various complex functions, which will be widely used for a long time.

## 5. Application

FMG, EMG, and EIT are three methods to obtain biological information of the human body, they can reflect different conditions of the human body. Therefore, they are widely used in human–machine interaction (HMI), medical, and healthcare. Some applications are shown in Figure 7. In this section, we will introduce some applications of three techniques in HMI, medical, and healthcare.

### 5.1. Human–Machine Interaction

#### 5.1.1. FMG in HMI

The earliest application of FMG can be traced back to the 1960s. In 1966, L. F. et al. proposed a device called the French electric hand, which uses a pneumatic pressure sensor to collect residual muscle motion signals of the amputee's forearm, and then controls the opening and closing of the gripper [121]. However, limited by the backward sensor manufacturing process and signal processing methods, FMG did not attract widespread attention at the time. 

In recent years, with the emergence of thin film piezoresistive sensors and machine learning algorithms, problems have been solved from both hardware and software, making it possible for FMG to perform more complex information recognition. The advancement of sensor manufacturing technology has made signal acquisition devices more portable and sensitive. The advancement of computer computing power has made deep learning possible. SVM, LDA, ANN, and other algorithms are also gradually applied to FMG. 

Nowadays, FMG is widely used in the field of artificial limb control. FMG directly detects the characteristics of muscle changes during exercise, so that it can better reflect the exercise situation. As the direct cause of movement, muscle contraction changes cause FMG to have a better signal-to-noise ratio than EMG, MMG, and other previous motion recognition methods. At the same time, it reflects the nature of movement more intuitively. Moreover, FMG has higher recognition accuracy and more accurate fatigue parameters than EMG in the case of high-speed motion [122]. Compared with EMG, FMG is less susceptible to sweat, motion artifact, electrode shift, and other electrical interference, and its devices are more affordable [29]. 

As early as 1999, researchers began to use FMG to control prostheses. Abboudi et al. fabricated pneumatic sensors using open-cell polymeric. They detect the movement of the tendon corresponding to specific fingers and control the movement of the prosthetic finger through a computer. They tested on three upper-limb amputees, and the result showed that voluntary flexions of individual fingers and grasping motions can be achieved after short training [26]. 

In 2001, their team proposed a linear filter using pressure vector decoding to decode the output signals of eight pneumatic sensors. They use pneumatic sensors to collect pressure signals around the residual muscles during finger movement, construct an eight-dimensional pressure characteristic matrix, and decode it to obtain finger movements [89]. However, limited by the data processing methods at the time, they could only recognize simple movements, such as finger taps, and their application was limited. 

Since 2006, FSR began to be used in FMG as a signal acquisition device [123,124]. In 2007, Ogris et al. used FSR to cover the lower and upper part of the forearm to capture arm muscle contractions caused by different gestures. They use C4.5 classifiers, *k*-nearest-neighbor (KNN), and hidden Markov models (HMM) to process the data separately. As a result, they proposed that, for all classifiers, the overall accuracy of the FSR system is between the accelerometer (between 5% and 10%) and the gyroscope (between 2% and 11%). Adding FSR to another sensor can increase accuracy by 1% to 29% [124]. 

Although Ogris et al. used classification algorithms, such as KNN, to process the data, the recognition accuracy they obtain was unsatisfactory and far from the application level. A few years later, with the popularization of more advanced algorithms, such as support vector machine (SVM) and linear discriminant analysis (LDA), the accuracy of gesture recognition has been greatly improved. 

In 2012, Li et al. designed a sensor array which is composed of 32 FSR sensors [7]. They use the SVM algorithm to recognize 17 hand movements including single-finger movement and multi-finger grabbing. Finally, they obtained an accuracy above 99% in the in-session validation. 

Since then, more and more teams have used different recognition algorithms to conduct research in different fields. In 2016, Ferrone et al. use a wristband equipped with stretchable strain gauge sensors to recognize 16 different gestures. They processed the signals by two machine learning algorithms (LDA and SVM) with an accuracy of 87% and 95%, respectively [125]. 

Another important area in prosthetic control is the estimation of force. In order to achieve precise control of the prosthesis, it is necessary to recognize the force used. Because FMG has the characteristics of intuitively reflecting muscle activity, it can reflect the magnitude of force by detecting the intensity of muscle activity to a certain extent. Sakr et al. use a total of 60 FSRs embedded in four bands, estimating in two cases: (1) 3-DoF force and 3-DoF torque at once and (2) 6-DoF force and torque [42]. The results showed that FMG achieves a good performance in multiple-DoF force/torque estimation.

#### 5.1.2. EMG in HMI

The earliest application of EMG can be traced back to the 1970s. As a quite mature technology, EMG has been widely used in the HMI area for the last few years. For example, in 2010, Wei et al. used EMG to control wheelchairs for the disabled. They collected forehead sEMG signals and facial information from six subjects. In the collection process, five facial expressions and movements were collected, corresponding to five commands to control the wheelchair. Both time domain and frequency domain parameters were used in feature extraction. The time-domain parameters were RMS, MAV, ZC, and WL. The frequency-domain parameters were the mean frequency of the signal (FNN) and median frequency of the signal (FMD). They adopted an SVM classifier. The final accuracy could reach 93.75%. Experiments show that this method can control the wheelchair accurately [126].

In 2019, Dwivedi et al. studied the continuous decoding of human movement, while previous research had mostly focused on decoding discrete human movements. They collected muscles from 16 sites of 11 subjects. In feature extraction, WL, RMS, and ZC parameters were used and they used a random forest algorithm for classification. The final classification result could reach 83.61%, verifying the feasibility of continuous action decoding [127].

In the same year, Sakib et al. proposed a prosthetic printing technology controlled by EMG. The prosthesis printing system mainly consists of an EMG recorder and prosthesis printing module. The printed materials are all locally sourced, reducing the cost of prosthetics for patients by 96 percent. They tested the printed prosthesis. They collected test information from 30 subjects and could achieve an accuracy of 87% [128].

In 2021, Choromański et al. used EMG signals to assess the muscle load of the driver to design a customized vehicle transition system for the user. They collected EMG from six muscles of the upper limbs of 30 male drivers aged 20–23 years. The experiment lasted about 5000 s. The sampling frequency was 1500 Hz, satisfying the sampling theory. This achievement solveed the problem that the disabled cannot control the steering wheel [1].

#### 5.1.3. EIT in HMI

The prototype of EIT can be traced back to the 1920s when some geologists proposed a resistivity imaging technology in order to study the mineral information of shallow geological distribution. They use an electrode array to measure the feedback voltage of current injected into the shallow surface, which is then analyzed and calculated to obtain information about the mineral deposits. As a relatively recent technique, EIT has been used in several fields, with many potentials yet to be explored. With the development of human–machine interaction technology, the application of EIT within it has also been widely studied.

The human wrist and arm have complex structures with bones and muscles. When people perform different gestures, the bones and muscles will show different states, which can be distinguished by the distribution of internal impedance. Therefore, EIT technology can provide a new method for gesture recognition. It only needs to wear a wearable bracelet on the wrist to realize accurate recognition by detecting the impedance distribution of the wrist. The gestures commonly used for testing include relaxing, fist, stretch, thumbs up, Spiderman gesture, pinch, and so on. Lots of teams have achieved some results in this regard.

Zhang et al. [3] designed a wearable hand ring called tomo based on 4-pole EIT to detect gestures, which has the advantages of having a light weight and low power consumption. It achieved high accuracy in determining the set of gestures. Russo et al. [129] experimented with contact location identification by using opposite drive patterns with 8 and 16 electrodes and performed a test for the control of the Kuka robot in real case scenarios by using recognition results. The results showed that this method realized a more accurate and faster recognition than traditional tomography approaches. More complex patterns are also proposed and researched. Usually, the measurement of voltage needs the adjacent electrodes. Since the distribution of human impedance is a three-dimensional problem, Jiang et al. [130] investigated the effect of using multilayer electrodes to measure three-dimensional EIT. The results show that 3D-EIT performs better than 2D in gesture recognition. Some application in human–machine interaction is showed in Figure 8. A summary of the different reference articles is shown in Table 4.

### 5.2. Medical and Healthcare

#### 5.2.1. FMG in Medical and Healthcare

Most of FMG’s application in the medical field focus on rehabilitation training and disease prediction. Neto et al. designed a tendon-actuated robotic glove with an optical fiber force myography sensor [47]. They propose a system that features tendon-driven actuation through servo motors, which can help users move their fingers according to wishes. The soft glove will associate with the recovery of people with hand disabilities. 

Quantitative evaluation of the gait phase can provide useful information for diagnosing gait abnormalities. Jiang et al. used a force myography-based technique to detect the gait phase, which can be used to diagnose gait abnormalities and specify a better rehabilitation plan to restore normal gait patterns [28]. They test this system to detect four different gait phases using an LDA algorithm. As a result, this approach can correctly detect more than 99.9% of gait phases, which has a promising potential in future application.

#### 5.2.2. Medical and Clinical Application of EMG

EMG signal is a good tool in many medical and clinical applications, for example, to evaluate physical condition, diagnose movement disorders, and assist in the rehabilitation of the disabled [143]. In this section, we will review the development of EMG in the medical and clinical fields.

As early as 1982, examples of using EMG for health monitoring have emerged. Jerrold et al. measured the frequency and amplitude components of the EMG signal during muscle contraction. They measured muscles in six males including the biceps, adductor pollicis, and quadriceps muscles [144]. They found that the high-frequency component of the EMG power spectrum was enhanced during fatigue contraction, and the linear relationship between EMG amplitude and tension was broken. This is an early example of researchers explaining EMG signals through TFD to detect human health.

In 2016, Buchner et al. used EMG to monitor patients’ muscle movements as they breathed. They used a new method called the blind source separation (BSS) algorithm in preprocessing, which effectively distinguished interference between other muscles, especially the muscles near the heart [145].

In 2018, Belfatto et al. used multi-domain analysis to evaluate post-stroke patients’ rehabilitation in robotic therapy. One of the parameters was EMG. They filtered EMG at an upper cut-off frequency of 50Hz and a lower cut-off frequency of 10 Hz and used a non-negative matrix factorization (NMF) algorithm. The sampling frequency was 1 kHz. Finally, the clinical improvement of the patients resulted in significant temporal and spatial changes in EMG signals [146].

It is worth noting that, in recent years, more and more wearable devices have been put into application, which is of great help to real-time health detection. In 2017, X. Li and Y. Sun presented a button-like wearable system for monitoring bio-electrical signals, such as EMG, ECG, etc. A rechargeable Li-ion battery was used to power the system. The whole system was integrated into a 39 mm × 32 mm × 17 mm little box. The entire system had a mass of just 24 grams [147].

In 2020, Zhao et al. made a wearable device for supervising upper limb rehabilitation conditions by combining ECG with EMG sensors. The raw ECG/EMG signals were processed and transmitted to mobile devices. Furthermore, they developed a software platform and realized the visualization of ECG/EMG information [148].

Some wearable devices that can provide medical-grade data are also becoming commercially available. Some companies, such as BioSemi instrumentation, Shimmer, and Biometrics Ltd., provide EMG sensors with lightweight, multi-channel, and high signal-to-noise ratios. Several new EMG instruments (e.g., FreeEMG) and EMG medical analysis kits (e.g., Biosignalssplux) have also been produced [143].

#### 5.2.3. Clinical Application of EIT

In recent decades, more and more attention has been paid to the application of EIT in medical human detection. In 1978, Henderson and Webster monitored lung ventilation using EIT to reconstruct cross-sectional images [149]. In 1983, Barber published the first electrical impedance tomography image to visualize the cross-section of a human forearm [150] and proposed the first EIT system for medical imaging in 1984 [151]. 

A typical application of EIT in biomedical imaging is thorax tomography. This method of measuring the thorax (especially the lungs) has received a lot of research. In clinical practice, pulmonary ventilation is often monitored to assess health or treatment status, so as to formulate a corresponding diagnosis and treatment plan. In 1995, Hahn et al. proposed functional electrical impedance tomography (f-EIT) to monitor regional ventilation in the lung by generating functional images of the electrical impedance distribution [152]. In 2003, Hinz et al. investigated the use of EIT to monitor the changes in the respiratory status of the lungs [153]. Improving the quality of imaging has a positive impact on formulating diagnostic protocols, so some investigators have suggested improvements to the traditional methods which improve the utility of EIT in the clinic. Hao et al. studied the influence of plane spacing of two planes of EIT sensors in lung imaging and proved that there was an optimal spacing [152]. In order to solve the problem of low spatial resolution of EIT, Li et al. proposed to combine the traditional CT technology with EIT. They developed a CT image-guided EIT (CEIT) and adopted cross gradient technology, to obtain a high-resolution image of lung detection [154]. 

When using ventilators to treat patients with acute respiratory distress syndrome (ARDS), some areas may collapse and others may overinflate due to the differences in ventilation status, which may cause damage to patients. Zhao et al. adopted EIT to quantify pulmonary improvement patterns as a global index of homogeneity (GI); thus setting a reasonable positive end expiratory pressure (PEEP) for the ventilator clinically, which has a protective implication for patients with ARDS [155,156]. Guillaume et al. studied the detection of PEEP indexes by EIT and accordingly provided a suitable ventilation strategy for extracorporeal membrane oxygenation (ECMO) [157]. At present, much of the literature has reported the results of using EIT to find the best PEEP [158,159]. 

In addition to detecting the ventilation, EIT can also be used for other pulmonary tests. Frerichs et al. proved the feasibility of EIT in detecting lung perfusion under the condition of using an electrical impact contrast agent [160]. Fagerberg et al. proposed the use of EIT to detect pulmonary perfusion and to identify the acute lung injury (ALI) caused by endotoxinaemic [161]. Pneumothorax is a noticeable complication during ventilation, and Costa et al. monitored the characteristic changes related to pneumothorax through EIT and developed an automatic detection algorithm [162]. Frerichs et al. also introduced EIT technology in the study of pulmonary ventilation in critically ill neonates and believed that it has bright prospects in the future [163]. Furthermore, EIT for asthma, ventilation weaning and expansion recoil, sequential lobar collapse, targeted physiotherapy, and pleural fluid assessment have also been intensively studied [164]. 

Thorax tomography is not the only application of EIT in biomedical imaging. Researchers have investigated methods for applying EIT to the brain, abdomen, and tumor detection. In 2001, Tidswell et al. used scalp electrodes to record the impedance changes of the human head for the first time [165]. Subsequently, Bagshaw et al. reconstructed impedance images of the human brain using FEM and suggested the possibility of applying it to epileptic treatment. Ayati et al. applied EIT to the in vitro localization and size estimation of intracranial hematomas. They compared the respective advantages and disadvantages of full array (FA) and semi array (SA) [166]. However, the low electrical conductivity of the skull reduces the penetration of the excitation current applied by EIT and distorts the brain resistance imaging. Methods to solve this problem currently remain to be investigated. EIT has also been used to perform gastrointestinal imaging to visualize gastric emptying and gastric secretion. There is a significant difference in the electrical impedance distribution between normal and tumor breast regions, demonstrating that EIT can be used to detect breast cancer [167]. Moreover, when treating tumors by hyperthermia (HT), there is a strict requirement for temperature control, whose change can be real-time reacted to by monitoring the impedance change of the treatment area through EIT; thus achieving a good temperature control effect [168]. Some application in medical is showed in Figure 9.

## 6. Summary and Comparison of the Three Techniques

In this section, we will summarize these three different technologies and compare their advantages and disadvantages.

### 6.1. FMG

In the early days, researchers adopted EMG for gesture recognition. On the one hand, due to the backwardness of the sensor manufacturing process, it is difficult to make a flexible and high sensitivity force sensor; on the other hand, the machine learning technology is not yet mature, which makes it difficult to link the movement and the pressure signal on the muscle surface. Nowadays, FMG has become an alternative to pattern recognition with higher accuracy than EMG [137].

Compared with EMG, FMG has the following advantages:

**1. Considerable output.** In the signal processing part, the FMG system often only uses a voltage divider bias circuit to obtain the required signal [9,17,136], while the EMG requires a series of operations, such as amplification and wavelet transform.

**2. Higher signal-to-noise ratio and anti-interference ability.** Piezoresistive force sensors directly reflect the relationship between pressure and resistance. Based on this principle, the relationship between pressure and voltage can be obtained. However, for EMG, the electrical signal fluctuates greatly, and the noise ratio is obvious. At the same time, FMG has good anti-interference ability. When faced with some daily scenarios, such as sweating and electromagnetic interference, the EMG signal will be greatly affected, while FMG is rarely affected, which means FMG has stronger robustness.

**3. Lower cost but similar recognition accuracy.** Compared to complex EMG systems, the most frequently used sensors in FMG systems is thin film piezoresistive sensors [29], for instance, FSR 402, FSR 400, and TPE 502C. Jiang et al. used FMG to build a gesture recognition system that only cost 1% of the traditional commercial medical-grade sEMG system, but obtained similar or better recognition accuracy compared to the EMG system [10].

**4. Better performance in dynamic motion.** In high-speed activity, FMG is considered to be a more suitable and accurate identification scheme because it can truly reflect the mechanical motion of human muscles at high speed [122]. However, EMG will be greatly affected due to high-speed movement.

## 6.2. EMG

EMG is the most commonly used biological signal when describing a muscle’s condition for a few decades. It is a traditional and easily accessible signal, which is quite mature and complete. Its signals are generated during the duration of the action, as a direct response to muscle movement, which means that it does not need outer stimulation. EMG has some characteristics as follows:

**1. EMG is generally proportional to the motion amplitude of the muscle.** Generally, when the muscle contracts stronger, the signal amplitude becomes stronger in accordance. 

**2. EMG amplitude is very small.** The amplitude of the EMG signal depends on the muscle condition, the type of exercise, and the observation condition. The amplitudes of the collected EMG signals are different for different people, different muscle types, and different movements, but they are all within a small range. For sEMG signals, its amplitude range is usually 0–10 mV [19,21,175].

**3. EMG precedes the muscle’s movement.** This is because the EMG signal is detected when the brain sends instructions to muscles to control movement.

## 6.3. EIT

With the rapid development of imaging technology in modern medicine, many perfect methods, such as computed tomography (CT), ultrasound imaging, and magnetic resonance imaging (MRI), have been developed. However, these traditional technologies and equipment are complex, expensive, and have their limitations. In contrast, EIT has different advantages.

**1. No radiation and damage.** Compared with the X-rays used in traditional X-CT, EIT injects the current with the frequency and amplitude within the safe range. Furthermore, nowadays the mainstream EIT uses non-invasive electrodes in human detection instead of the plug-in electrodes used in traditional EIT and will not cause harm to the human body.

**2. High-time resolution.** EIT imaging is rapid and sensitive to the impedance changes caused by physiological components and state changes. Compared with CT and MRI, it has better performance in reflecting the physiological activities of the measured parts in real-time.

**3. Simple and portable device.** EIT does not need large transmitting and receiving devices, but only small electrodes. It just needs small volume, low-cost devices, and has no special requirements for the working environment.

## 7. Challenge

Despite their various advantages, they still have some problems to solve. In this section, we will analyze the current problems and challenges faced by these three technologies. The comparison of the three technologies is shown in Table 5.

### 7.1. FMG

**1. Preload error.** The performance of the force sensors utilized by FMG is largely related to the installation position and tightness, which are often not consistent during installation. This results in a different preload force for each installation process. Although some errors can be reduced by standardization, such errors still exist and affect the reliability of the data.

**2. Sensor shift.** Because the sensor cannot be fixed exactly in a certain position, the sensor will inevitably shift during vigorous exercise. Although it will only cause small displacements, when the displacements continue to accumulate the sensors may shift, largely worsening the reliability of the signals.

### 7.2. EMG

**1. Powerline interference.** The frequency of EMG main energy concentrates in 20–150 Hz, and its amplitude is quite small. Therefore, noises, such as 50 Hz working frequency interference, high-power equipment interference, and background noise, are not conducive to acquiring EMG signals. Secondly, the electrode noise also needs to be considered, which is related to the electrostatic interference and the mechanical movement between the electrode and the skin. The shape of the electrode and the improper placement might produce noise that cannot be ignored [81].

**2. Interference of skin surface factors.** It is mainly aimed at sEMG. When sampling human skin surface, skin surface condition has a great influence. For example, sweat will affect the conductivity of the electrode, leading to changes in the impedance between skin and electrode [5]. The hair on the skin surface and the fat under the skin will also affect the impedance. To make good contact between the electrode and the skin, pressing can be adopted, but pressing the electrode will leave pressing marks on the skin surface, which will affect the impedance as well. In addition, the electrodes may irritate the skin, causing unstable signal acquisition. 

**3. Motion artifacts.** Motion artifacts are generated by the muscles tested in different types of contraction, which will pollute the low-frequency components of EMG and lead to signal distortion [176,177]. Many factors must be considered, including the type of muscle and contraction being tested, sensor configuration, and the specific noise source. Eliminating motion artifacts needs to determine the appropriate filter specification.

**4. Natural frequency instability of EMG signal.** The amplitude of the EMG signal is random, especially in the frequency range of 0–20 Hz, and most components are unstable [20].

### 7.3. EIT

**1. The spatial resolution of EIT is relatively low**, especially in the areas close to the center. This seriously affects the application of EIT. At present, some researchers combine EIT with CT or MRI to obtain CT image-guided EIT (CEIT) and magnetic resonance electrical impedance tomography (MREIT) technology, attempting to improve the resolution of EIT in this way.

**2. Complicated inverse problem.** Because of the illness, non-linearity, and uncertainty of the inverse problem, the algorithm for solving is often computationally intensive and the results of EIT are not accurate enough to reduce the resolution and affect the imaging time.

**3. The 3D characteristics of EIT.** The traditional EIT techniques all consider electrical impedance distribution as a two-dimensional problem, based on which electrodes are placed in a plane and assuming that the current flows only through the plane to be measured, which reduces the measurement and calculation difficulties, but in fact, the distribution of electrical impedance is a complex three-dimensional problem, which needs further study.

**4. Quantification of EIT results.** Currently, there is still a lack of technology to quantify EIT, which hinders its application.

## 8. Outlook

The above sections explain the working principles, device architectures, interpretation algorithms, practical applications and challenges of EMG-, FMG-, and EIT-based human body signal detection techniques. Based on the content discussed, the authors believe that these three technologies have the following development trends in the foreseeable future.

### 8.1. Complementation and Calibration

From this article, the three technologies reflect the same thing from different perspectives and they are correlated closely. Therefore, the fusion of these three biological signals can compensate for each other’s shortage, which is of great help to improve the performance of the system [8]. For instance, impedance changes strongly impact the EMG; hence EIT could be used to retrieve the impedance profile to calibrate the EMG signal.

### 8.2. Broaden Application Scenarios

Wearables are powerful in monitoring human body signals omnipresently. Therefore, the three technologies reviewed in this article have the potential to be used in telemedicine. By collecting the patient’s biosignals, doctors can obtain the patient’s biological indicators in real-time and provide guidance for the patient’s medical rehabilitation. Secondly, the fusion of various biological signals makes it possible to build a digital twin model of the human body [178]. The digital twin model allows doctors to study the impact of different drugs and different rehabilitation strategies on the patient’s body from the patient’s digital model, and then choose the best solution for the patient. The biggest advantage of the digital twin is that it does not affect real individuals, and all work is carried out in the virtual world, which will be the direction of research in the future.

## Figures and Tables

**Figure 1 biosensors-12-00516-f001:**
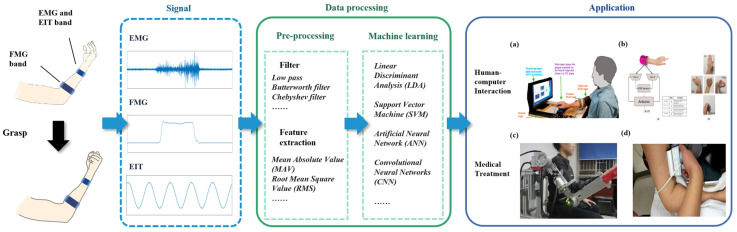
The generation, processing, and application of FMG, EMG, and EIT signals. (**a**) Use FMG to predict forces in two directions [12]; (**b**) a novel kirigami-based bracelet senses the skin impedance signals, which is used to distinguish between different gestures [13]; (**c**) identify the movement intention based on sEMG [14]; (**d**) an EIT-based technique for assessing spinal cord injury [15].

**Figure 2 biosensors-12-00516-f002:**
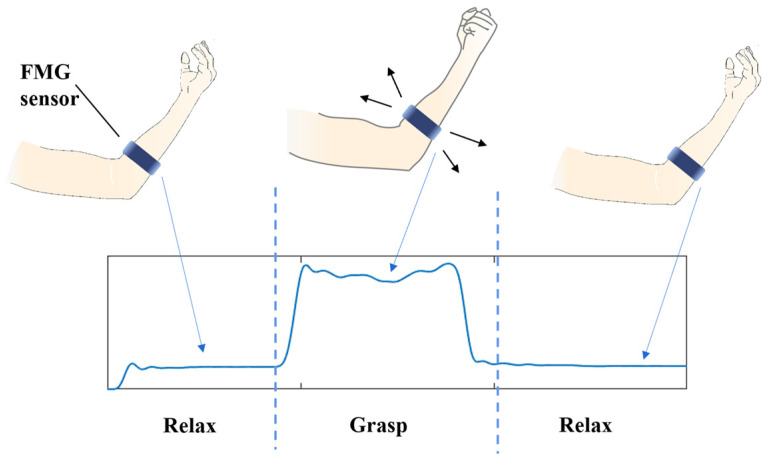
Single FMG sensor output signal during the relax–grasp–relax process.

**Figure 3 biosensors-12-00516-f003:**
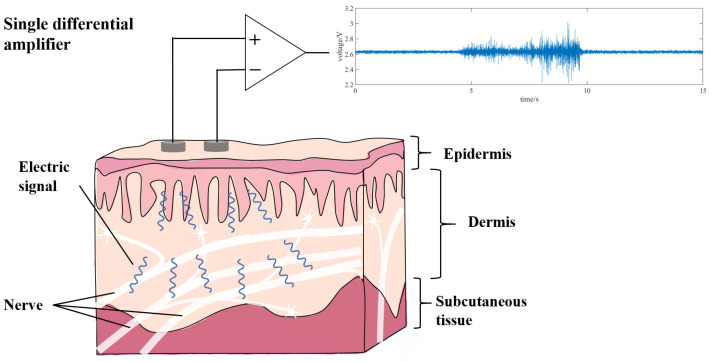
EMG refers to a series of electrical signals associated with muscles due to neurological control and generated during muscle contraction.

**Figure 4 biosensors-12-00516-f004:**
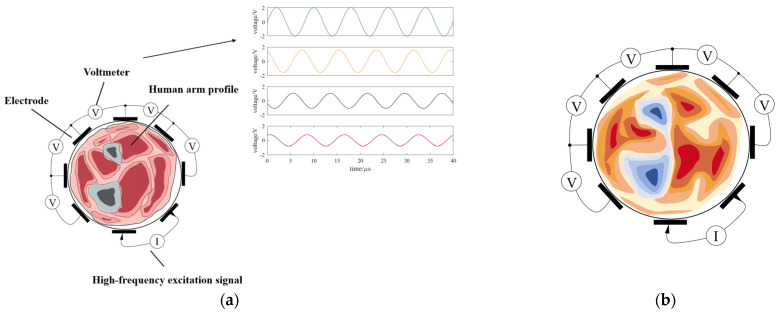
(**a**) EIT electrode distribution and four−channel voltage signal under high−frequency excitation; (**b**) impedance map reconstructed using the voltage signal acquired from (**a**), where the redder the color, the larger the impedance.

**Figure 5 biosensors-12-00516-f005:**
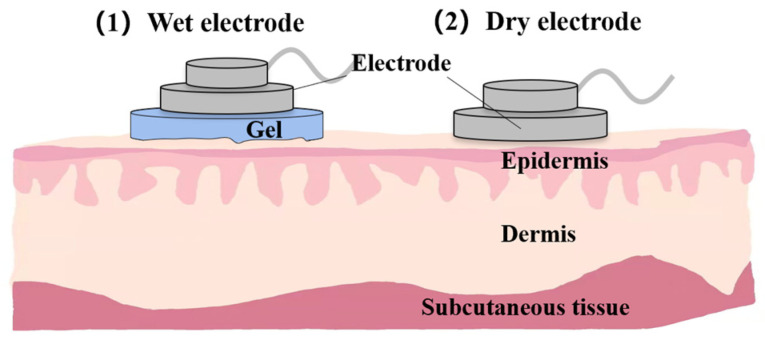
Wet electrode (**right**) and dry electrode (**left**).

**Figure 6 biosensors-12-00516-f006:**
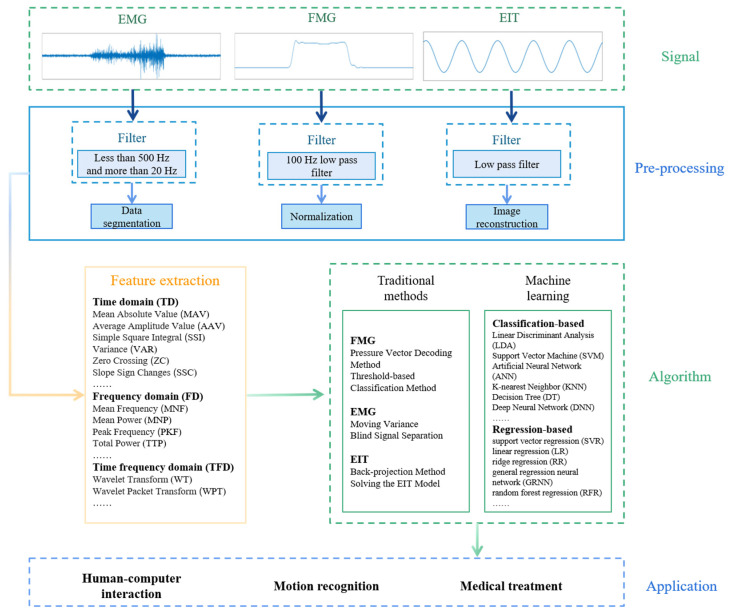
All steps of data processing.

**Figure 7 biosensors-12-00516-f007:**
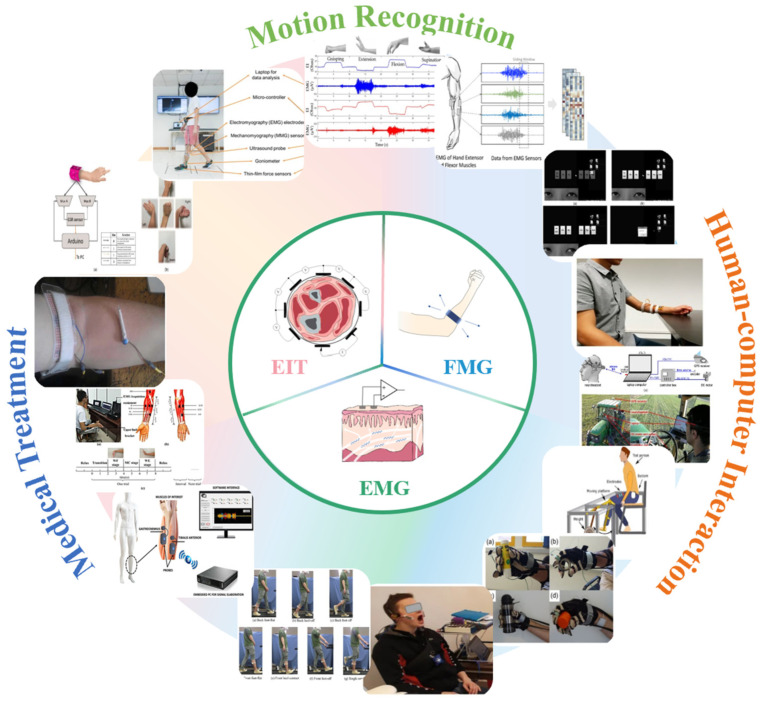
Application of FMG, EMG and EIT. From the top center picture, in a clockwise order: bionic control based on EIT, EMG and FMG [109]. An EMG gesture recognition system [110]. Human–machine interaction system based on EOG and temporalis EMG [111]. Feature optimization of sEMG in human–machine interaction [112]. Tractor manipulation via EMG-based human–machine interface [113]. Combining synchronized EMG and EIT to measure muscle activity [114]. A disabled assistive robotic glove using optical fiber force myography sensor [47]. Differential diagnosis of temporomandibular joint disorders using sEMG [115]. Gait Phase Detection [116]. Evaluation of sarcopenia based on sEMG platform [117]. Time-frequency muscle synergy estimation based on sEMG [118]. Peripheral blood vessel puncture control system based on electrical impedance measurement [119]. A novel kirigami-based bracelet is used to sense the skin impedance signals for distinguishing between different gestures [13]. A method based on EMG, MMG, and ultrasound images to study internal muscle morphological changes in stroke survivors while walking [120].

**Figure 8 biosensors-12-00516-f008:**
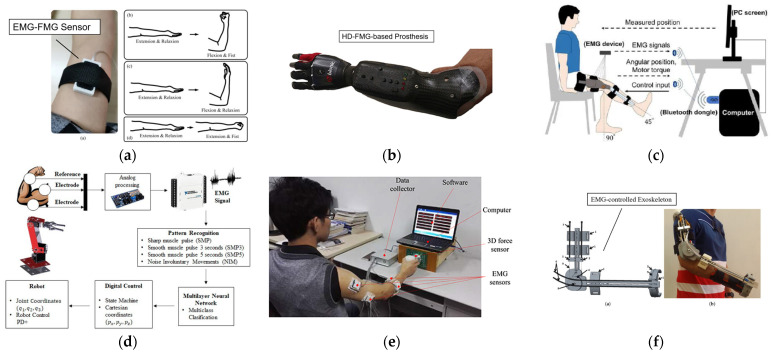
Application in human–machine interaction. (**a**) A myoelectric prosthesis control based on the combination of EMG and FMG [5]; (**b**) a prosthetic control using high-density FMG [131]; (**c**) an EMG-controlled dynamic model for musculoskeletal simulation and exoskeleton control [132]; (**d**) application of EMG pattern recognition in manipulator control [133]; (**e**) estimation of grip strength and three-dimensional push–pull force using electromyography [134]; (**f**) a control scheme of the elbow joint memory alloy exoskeleton based on sEMG signals [135].

**Figure 9 biosensors-12-00516-f009:**
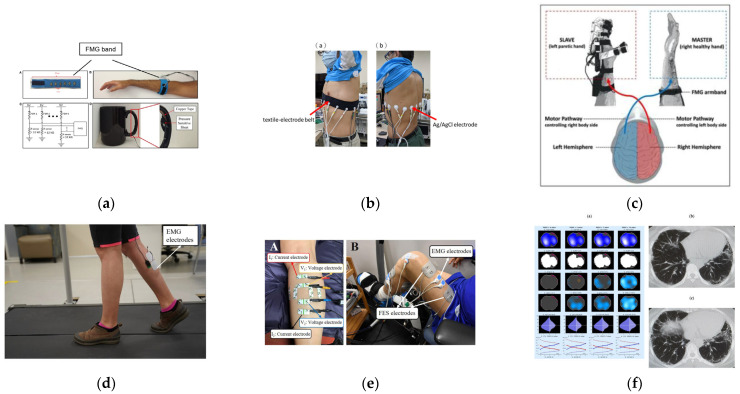
Application in medical. (**a**) Use FMG and machine learning techniques to differentiate between grasping and no grasping [169]; (**b**) textile electrodes integrated with a clothing belt for EIT lung imaging [170]; (**c**) assisted rehabilitation design of 3D printed hand exoskeleton based on FMG control [171]; (**d**) EMG biofeedback device for gait rehabilitation [172]; (**e**) detection of changes in lower extremity muscle impedance properties immediately after functional electrical stimulation-assisted cycling training in chronic stroke survivors [173]; (**f**) an evaluation of spontaneous respiratory idiopathic pulmonary fibrosis using EIT [174].

**Table 1 biosensors-12-00516-t001:** Summary of sensor techniques in FMG.

Mechanism	Material	Measuring Range	Hysteresis	Advantage	Disadvantage
Piezoresistive [27,28,29,30] 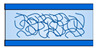	PSS film, PEI film, Acrylic, polyester	0.2–20 N	10%	Thin and flexible, simple and easy to integrate, convenient and affordable, widely application	High power consumption, high hysteresis, low sensitivity, and temperature drift
Capacitive [23,34,35,36,43,44] 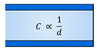	Silicon, PDMS, SiO_2_, PET, Au	0–20 N	7–35%	Low power consumption, simple structure, high resolution, high sensitivity	Sensitive to EMI noise, susceptible to heat and moisture
Piezoelectric [24,45,46] 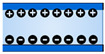	PP film, PVDF film, PET, PEN	0.5–40 N	3–5%	Light weight, stretch ability, strong sensitivity, low power consumption, suitable for dynamic application	Cannot measure static forces, susceptible to heating
Optical [2,25,47,48] 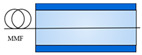	PVC plates, graphite, silica multimode fiber	0–10 N	6.3–20%	Immune to EMI, low cost, smaller linear errors, lower delays	Complex architecture, high power consumption, low spatial resolution

**Table 2 biosensors-12-00516-t002:** Different electrode types.

Electrode	Materials	Means of Reducing Contact Impedance	Electrode–Skin Equivalent Model	Advantages	Disadvantages	Application Scenarios
Dry electrode	Gold-plated silk fabrics, such as silvered yarn [50], silvered nylon [51], etc.	Using Hydrogel membrane or saline moisturizing interface [52]	Complex (the coupling of other interference) [53]	Contactless. Simple measuring conditions. Little stimulation to human skin. Low cost. Suitable for long-term measurement	Difficult to attach to the skin. The accuracy of measurement is worse	Wearable devices for long-term use
Wet electrode	Metal mixtures, such as Ag/AgCl [54,55], aluminum, gold/gold chloride [56], etc.	Using a wet gel layer	Simple (Containing double-layer capacitors, parallel or series resistors) [53]	Easy to attach to human skin [53]. Simple structure. Suitable for short-term measurement	Performance decreases over time. Human skin will be stimulated	Clinical care. Short-term health monitoring

**Table 3 biosensors-12-00516-t003:** Some commonly used features in TD, FD, and TFD.

Domain	Parameter	Concrete Explanations	Abbreviation
TD	Average Amplitude Value	The average amplitude of the signal	AAV
	Mean Absolute Value	/	MAV
	Simple Square Integral	signal energy	SSI
	Variance	/	VAR
	Zero-Crossing(s)	The number of times the signal waveform intersects the axis “0”	ZC(S)
	Slope Sign Changes	Change in the sign of the slope	SSC
	Waveform Length	/	WL
	Root Mean Square Value	/	RMS
FD	Mean Frequency	/	MNF
	Mean Power	/	MNP
	Peak Frequency	Maximum frequency	PKF
	Total Power	/	TTP
	Power Spectral Density	/	PSD
	Power Spectrum Ratio	/	PSR
TFD	Wavelet Transform	/	WT
	Wavelet Packet Transform	/	WPT
	Short-Time Fourier Transform	/	STFT

**Table 4 biosensors-12-00516-t004:** Summary of application in HMI.

Method	Reference	Sensors Number	Sampling Frequency	Feature Extraction	Algorithm	Function	Performance
FMG	[136]	16	15 Hz	MAV	LR, SVR, NNR, and RF LDA SVM	Predict the angle between index finger and thumb (θ_TI_), the angle between middle finger and thumb (θ_TM_)	A correlation of determination (R^2^) of 0.871 for θ_TI_ and 0.941 for θ_TM_
	[22]	64	10 Hz	Mean absolute value slope	LDA	Distinguish 11 gestures in static and dynamic conditions	Accuracy over 99% in static conditions, and accuracy over 86% in dynamic conditions
	[137]	384	15 Hz	MAC	SVM	Propose a proportional control method to classify six gestures	Classification accuracy of 83.4 ± 3.47%
	[27]	12	/	Mean absolute value slope	Threshold-based classification method	Detect six hand motions intention and estimate grasping force	Average accuracy of 98 ± 1.3% on six subjects, implement a proportional force control
	[17]	2	1 kHz	MAV, RMS, MAX, SUM	Fuzzy logic-based classification scheme	An affordable hand prosthesis to distinguish six different grip patterns	An offline accuracy of 97 % on thirteen subjects
	[89]	8	200 Hz	RMS	Pressure vector decoding	Provide biomimetic finger control	Successfully controlled flexion of three phantom fingers
	[16]	8	25 Hz	PSD, likelihood	RNN	Develop an effective human–robot collaboration scheme	Estimate human intentions in <1 s and decide to assist or avoid the human body
	[46]	8	10 Hz	MAV	KNN	Propose a step counter to detect low-speed walking steps (<2.2 km/h)	A low error rate (<1.5%) at three walking speeds
EMG	[82]	4	1024 Hz	Integral of Absolute Value, VAR.	GK-SVM with PE or Wilson Amplitude (WAMP)	Distinguish gestures of standing, squatting, and sitting, upstairs, downstairs, and walking	Seven kinds of ADLs and falls were classified with accuracy from 96.43% to 97.35%
	[84]	8	200 Hz	MAV, ZC, SSC and WL.	CNN	Distinguish open hand, closed hand, wrist extension, wrist flexion, ulnar deviation, and radial deviation	Average accuracy of 97.81% on a database of seven hand and wrist gestures
	[138]	5	2 kHz	CNN, RNN, Flourier Transformation.	Recurrent convolutional neural networks (RCNNs)	Distinguish five motions: biceps brachii, triceps brachii, anterior deltoid, posterior deltoid, and middle deltoid	An accuracy of 86.5–94.7% on eight subjects from two data sessions
	[83]	8	2 kHz	WL, MAV, WAMP, Cardinality (CARD), SSC and ZC	LDA	Distinguish nine hand gestures	An accuracy of 84.78–98.56% on nine hand gestures of eight participants
	[85]	4	1 kHz	MAV, ZC, WL and SSC	SVM	Distinguish six-foot movement: lift the toe, lift the heel, move the toe to the right, move the toe to the left, lean on the heel, lean on the toe, and rest foot	An accuracy of 52.86–95.71 for one channel; 81.43%-almost 100% for four channels
	[139]	12	2 kHz	MAV, VAR, MAV slope (MAVSLP), and WL	Convolutional neural network–long short-term memory network (CNN-LSTM)	Distinguish gestures in EMG signal dataset Ninapro DB2	The accuracy of 17 gestures is 83.91%. The accuracy for 20 subjects is 99.17%
	[140]	4	20 Hz	Multivariate Multiscale Entropy (MMSE) and Multivariate Multiscale Fuzzy Entropy (MMFE)	SVM	Data of uterine EMG	An accuracy of 86.4–96.5% on 300 records of the TPEHG DB database.
EIT	[88]	8	40 kHz	/	SVM, RF, KNN, LR, Adaboost	Worn on the wrist to classify 11 gestures with different algorithms	Accuracy is higher than 95%, in the Adaboost algorithm achieved the highest accuracy of 98.11%
	[3]	8	40 kHz	/	SVM	Test the accuracy of hand set with seven gestures and pinch set with four gestures on wrist and arm, respectively	Achieved higher accuracy on the wrist than on the arm, with the highest accuracy of 96.6%
	[141]	8, 16, 32	40 kHz	/	SVM	Test the accuracy with different electrode numbers for 11 gestures	Get an accuracy of 88.5% with 8 electrodes, 92.4% with 16 electrodes, and 94.3% with 32 electrodes
	[130]	16	125 kHz	/	DT (Fine Tree, Medium Tree), SV (Quadratic, Cubic, Medium Gaussian), ANN	Using different algorithms to test the accuracy of 2D and 3D EIT with different wristband separations	96.6% for DT(Cubic), 97.4% for (Medium Gaussian), and 97.7% for ANN, 5cm band separation is the best
	[142]	8	40 kHz	/	SoftMax, SVM, CNN	Worn on the forearm to classify 10 gestures with different algorithms	CNN has the highest accuracy of 96.66% for all the 10 gestures
	[64]	8	50 kHz–1 MHz	/	SVM	Worn on the wrist to classify three gestures with four different electrode materials	An accuracy of 76.7% with medical electrodes, 93.3% with conductive cloth electrodes, 96.7% with conductive cloth electrodes, 96.7% with curved copper electrodes
	[73]	8	20 kHz	/	Quadratic Discriminant	Test the accuracy for nine gestures based on two-terminal EIT	Obtain accuracy of 98.5%

Because EIT uses the magnitude as a feature, the feature column is empty.

**Table 5 biosensors-12-00516-t005:** Comparison of three techniques.

Technique	Robustness	SNR	System Complexity	Frequency	Cost	Advantage	Disadvantage
FMG	Excellent	High	Simple	0–100 Hz	Low	Considerable output, high anti-interference ability. Better performance in dynamic motion. Suitable for most situations	It is difficult to ensure that the sensors are exactly installed in the same location and have the same pressure. Sensors may shift during use
EMG	Poor	Low	Normal	20–500 Hz	High	Signal ahead of action, better predictability	Equipment noise. Interference of skin surface factors. Motion artifacts and natural frequency instability
EIT	Poor	Low	Normal	1 k–1 MHz	Low	Reflect the internal physiological state of the detection area	Low spatial resolution. Complicated inverse problem. The results are difficult to quantify

## Data Availability

Not applicable.

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
