# Peer review of "A Review of EMG-, FMG-, and EIT-Based Biosensors and Relevant Human–Machine Interactivities and Biomedical Applications"

_biosensors, 2022, doi:10.3390/bios12070516_

Round 1

Reviewer 1 Report

This article presents three techniques - Electromyography (EMG), Force Myography (FMG) and Electrical Impedance Tomography (EIT), applicable in Human-Machine Interaction (HMI). The idea of the authors is through this review to help readers to get acquainted in detail with the principles of operation, device architectures, interpretation algorithms, application examples, advantages and disadvantages and obstacles that need to be overcome. For this purpose, a detailed literature study was conducted, including 176 reference sources for a period covering the last 35 years. The oldest sources are related to the principles of the respective measurements, and those from the last decade reflect the current trends and achievements, as well as the directions for future development. In general, well-known facts are presented in the separate sections, but there are no in-depth and professional comments, and some of the statements and conclusions are inaccurate.

The title and the abstract correctly present the content of the article.

Remarks, questions and comments:

1.      Ln 262 - Following the logic of the explanation, it should be the “highest”, not the “lowest” frequency.

2.      Ln 303 - An electrode that has direct electrical (galvanic) contact with the body is considered contact, regardless of whether a contact gel is used.

3.      Ln 322 - Ln 328 - The relationship between the number of muscles, the number of channels and the sampling frequency is misinterpreted. First, when applying sEMG, the registered signal is a superposition of the electrical activity (electrical potentials) of all muscles in the area of ​​signal registration, and is not necessary to increase the number of channels. Second, the sampling frequency is chosen relative to the highest frequency in the spectrum of the recorded sEMG signal, and all multi-channel acquisition circuits/systems allow synchronous sampling of all channels so that no phase error is introduced.

4.      Ln 333 - How is defined the term "bipolar electrodes"?

5.      Ln 373 - More serious argumentation of the specified frequency range is needed. What are the values of the applied current?

6.      Ln 975 – Ln 977 - The levels of the amplitude of the EMG signal are not correct and do not correspond to those in the cited literature source.

Ln 1014 - The correct term is “Powerline interference”. 

Author Response

Question 1: Following the logic of the explanation, it should be the “highest”, not the “lowest” frequency

Reply: Thank you for this comment. The meaning in the sentence is to reduce the system cost by finding the highest frequency that can reflect all the motion information of the human body. Therefore, "highest" should be used in this place, "lowest" has been changed to "highest" in the article

Question 2: An electrode that has direct electrical (galvanic) contact with the body is considered contact, regardless of whether a contact gel is used.

Reply: Thank you for this comment. We recognize that the use of "contact" and "non-contact" is inaccurate. Therefore, in the article we use "wet electrodes" instead of "contact electrodes" and "dry electrodes" instead of "non-contact electrodes"

Question 3: The relationship between the number of muscles, the number of channels and the sampling frequency is misinterpreted. First, when applying sEMG, the registered signal is a superposition of the electrical activity (electrical potentials) of all muscles in the area of signal registration, and is not necessary to increase the number of channels. Second, the sampling frequency is chosen relative to the highest frequency in the spectrum of the recorded sEMG signal, and all multi-channel acquisition circuits/systems allow synchronous sampling of all channels so that no phase error is introduced.

Reply: Thank you for this comment. This is a very important point. We have clarified the relationship between the number of channels and the sampling frequency of movement in EMG signal process. Relevant modifications are as follows.

For the number of channels, when the action is simple, the sensor with few channels can be used. For complex movements or the combination of many single movements, it is appropriate to use multi-channel electrodes. Generally, the sampling frequency of each channel is usually the highest frequency of the recorded EMG signal to satisfy the Shannon sampling theorem. At the same time, the multi-channel acquisition system samples all channels synchronously, so that no phase error is introduced.

Question 4: How is defined the term "bipolar electrodes"?

Reply: Thank you for this comment. After reading the whole article, we found that “bipolar electrodes” did not contribute to the article. Therefore, to disambiguate, we remove “bipolar electrodes” from the article.

Question 5: More serious argumentation of the specified frequency range is needed. What are the values of the applied current?

Reply: Thank you for this comment. We researched more papers and add the information of frequency and value of current into this paragraph. The details are as follows:

The most common way is to use the current drive and voltage measurement nowadays. When the device works, the electrodes inject excitation current in turn, with a frequency in the beta range (10 kHz -1 MHz) which has been proved to be best suited for the measurement of tissue impedance. Then other certain electrodes measure the voltage response and pass the data to the backend. In modern EIT systems, the frequency of injection current is usually varied in a range between 50 kHz and 250 kHz. The maximal root mean square of the injection current is regulated in the standard IEC 60601-1:

Question 6: The levels of the amplitude of the EMG signal are not correct and do not correspond to those in the cited literature source. The correct term is “Powerline interference”.

Reply: Thank you for this comment. Because the magnitude of EMG is largely dependent on observation, we have revised this paragraph as follows after reviewing the literature. At the same time, we change “Equipment noise” to “Powerline interference”.

The amplitude of the EMG signal depends on the muscle condition, the type of exercise, and the observation condition. The amplitudes of the collected EMG signals are different for different people, different muscle types and different movements, but they are all within a small range. For sEMG signals, its amplitude range is usually 0-10 mV [19,21,175].

Reviewer 2 Report

The Authors introduced a comprehensive review of EMG, FMG and EIT biosensors for wearable devices to be used in HMI and medical applications. The work is interesting and useful, as it provides a good synthesis of principles, signal processing techniques, application scenarios, advantages, and disadvantages of these three technologies, that may be of great help to academic or industrial researchers approaching such topics.
However, I have noticed a few flaws and missings that need to be addressed.

Figures 3 and 6: In the representation of skin layers, "genuine leather" is not appropriate. Please refer to any manual of anatomy.

Moreover, Figure 3, legend: description of how EMG signal is propagated is misleading: actually, the signal is not propagated through nerves, up to the electrodes, but signals are generated by muscle fibers and propagated thanks to the conductivity of the surrounding tissues.

Line 321: Channel number and sampling frequency are, more correctly, the key indexes to measure the quality of EMG acquisition systems, together with noise rejection performances.

Lines 322-326: it sounds like sampling frequency is proportional to the number of channels. This is not true, in general, unless a single ADC with multiplexing is used for all the channels. In practice, each channel has its own ADC on most clinical grade systems. Basically, one may have a single channel and require a high sampling frequency: it depends on the dynamics he or she wants to detect in the acquired signal. If the goal is to detect tremor or on-off simple events, a low sampling frequency (50-100Hz) is enough; if we want to distinguish between voluntary contraction and myoclonus, a far higher sampling frequency is needed.

The following EMG-based wearable device (MYO armband) is worth citing for HMI applications DOI:10.1145/2968120.2987731
And this EMG-based wearable device is also worth citing for medical diagnostic applications.

Last, there are several incomplete or mistyped sentences. E.G.: lines 27, 34, 106,113, 116, 140-143, 194-195, 200, 372, 509-510, 894, 968-969, etc. A thorough check of English language is required throughout the whole document.

Author Response

Question 1: In the representation of skin layers, "genuine leather" is not appropriate. Please refer to any manual of anatomy.

Reply: Thank you for this comment. We changed “genuine leather” to “Dermis” after referring to a manual of anatomy in figure 3 and 6.

Question 2: The signal is not propagated through nerves, up to the electrodes, but signals are generated by muscle fibers and propagated thanks to the conductivity of the surrounding tissues.

Reply: Thank you for this comment. After our inspection, we found that the legend was wrong, but the description of EMG generation in the text is correct, so we changed the original legend and picture as follows:

Figure 3. EMG refers to a series of electrical signals associated with muscles due to neuro-logical control and generated during muscle contraction.

Question 3: Channel number and sampling frequency are, more correctly, the key indexes to measure the quality of EMG acquisition systems, together with noise rejection performances.

Reply: Thank you for this comment. This is a very important point. We make the following changes to the original expression:

Channel number, sampling frequency, and noise rejection performances are the key indexes to measure the quality of EMG acquisition system.

Question 4: it sounds like sampling frequency is proportional to the number of channels. This is not true, in general, unless a single ADC with multiplexing is used for all the channels. In practice, each channel has its own ADC on most clinical grade systems. Basically, one may have a single channel and require a high sampling frequency: it depends on the dynamics he or she wants to detect in the acquired signal. If the goal is to detect tremor or on-off simple events, a low sampling frequency (50-100Hz) is enough; if we want to distinguish between voluntary contraction and myoclonus, a far higher sampling frequency is needed.

Reply: Thank you for this comment. This is a very important point. We have clarified the relationship between the number of channels and the sampling frequency of movement in EMG signal process. Relevant modifications are as follows.

For the number of channels, when the action is simple, the sensor with few channels can be used. For complex movements or the combination of many single movements, it is appropriate to use multi-channel electrodes. For sampling frequency, when we just need to detect same simple event like tremor or any on-off event, a low sampling frequency can accomplish the task. However, if we need to focus on the details of one movement in order to better separate it from the others, we need a high sampling frequency in that case. Generally, the sampling frequency of each channel is usually the highest frequency of the recorded EMG signal to satisfy the Shannon sampling theorem.

Question 5: A thorough check of English language is required throughout the whole document.

Reply: Thank you for this comment. We recognize that language does have huge problems. Therefore, we submitted the manuscript to the MDPI English Editing Service and have received.

Round 2

Reviewer 1 Report

The corrections made are adequate.